# Pulse Protein Isolates as Competitive Food Ingredients: Origin, Composition, Functionalities, and the State-of-the-Art Manufacturing

**DOI:** 10.3390/foods13010006

**Published:** 2023-12-19

**Authors:** Xiangwei Zhu, Xueyin Li, Xiangyu Liu, Jingfang Li, Xin-An Zeng, Yonghui Li, Yue Yuan, Yong-Xin Teng

**Affiliations:** 1National “111” Center for Cellular Regulation and Molecular Pharmaceutics, Key Laboratory of Fermentation Engineering (Ministry of Education), Hubei Key Laboratory of Industrial Microbiology, Hubei University of Technology, Wuhan 430068, China; xiangwei@ksu.edu (X.Z.);; 2Department of Grain Science and Industry, Kansas State University, Manhattan, KS 66506, USA; yonghui@ksu.edu; 3School of Food Science and Engineering, South China University of Technology, Guangzhou 510641, China; xazeng@scut.edu.cn; 4Center for Nanophase Materials Sciences, Oak Ridge National Laboratory, Oak Ridge, TN 37830, USA; yuany@ornl.gov

**Keywords:** pulse protein, composition, structure–property relationship, functional property, physical modification, non-covalent complexation, food application

## Abstract

The ever-increasing world population and environmental stress are leading to surging demand for nutrient-rich food products with cleaner labeling and improved sustainability. Plant proteins, accordingly, are gaining enormous popularity compared with counterpart animal proteins in the food industry. While conventional plant protein sources, such as wheat and soy, cause concerns about their allergenicity, peas, beans, chickpeas, lentils, and other pulses are becoming important staples owing to their agronomic and nutritional benefits. However, the utilization of pulse proteins is still limited due to unclear pulse protein characteristics and the challenges of characterizing them from extensively diverse varieties within pulse crops. To address these challenges, the origins and compositions of pulse crops were first introduced, while an overarching description of pulse protein physiochemical properties, e.g., interfacial properties, aggregation behavior, solubility, etc., are presented. For further enhanced functionalities, appropriate modifications (including chemical, physical, and enzymatic treatment) are necessary. Among them, non-covalent complexation and enzymatic strategies are especially preferable during the value-added processing of clean-label pulse proteins for specific focus. This comprehensive review aims to provide an in-depth understanding of the interrelationships between the composition, structure, functional characteristics, and advanced modification strategies of pulse proteins, which is a pillar of high-performance pulse protein in future food manufacturing.

## 1. Introduction

By 2050, the world’s population will exponentially increase to over 10 billion from the current 7.9 billion, according to the World Health Organization (UNFPA, 2020). A major challenge of food scarcity will arise from climate change, the rapid growth of the global population, and imbalance in food production, which may inevitably lead to severe human malnutrition. Protein-energy malnutrition (PEM) is responsible for six million deaths worldwide annually [1]. The main source of dietary protein is highly reliant on animal-derived products, such as muscles, eggs, dairy, and their processed products, although livestock farming generates more pollution including sewage and greenhouse gas than crop production [2,3,4]. For instance, Heller and Keoleian compared the environmental burdens of beef production with the plant-based meat analog from Beyond Meat and they found that beef production results in 47–99% more energy, land, and water consumption plus 89% more greenhouse gas emissions [5]. Similar studies by Poore and Nemecek also confirmed that the resource consumption of plant-based dairies was significantly lower than the real ones [6,7]. Therefore, increasing the production and consumption of plant protein can be one of the potential solutions for addressing the challenges in sustainable agricultural and food development.

Pulse crops have drawn increasing attention in the food industry due to their low production cost, non-GMO status, and high yield of nutritious proteins [8]. According to the Food and Agriculture Organization (FAO) of the United Nations in 2023, pulses, the seeds of leguminous plants, serve 36 food and feed purposes and offer benefits to both food security and environmental sustainability. In terms of nutritional composition, pulse seeds contain more than 30% protein, carbohydrates including digestible and resistant starch, and dietary fibers, as well as essential vitamins and minerals and bioactive phytochemicals [9]. For these reasons, pulses have been suggested as wholesome alternatives to animal proteins. With their advantages of hypoallergenicity, broad acceptance, and better bioaccessibility, pulse proteins have gained popularity in the supply chain [10]. However, pulse crops, unlike common staple crops (e.g., soybeans), originate from a vast array of sources and showcase a remarkable diversity of species [11,12]. As a result, pulse proteins derived from different crops are distinguished from each other in structure, composition, and especially in functional properties. To achieve the full potential of pulse proteins in food applications, it is important to gain a comprehensive understanding of primary pulse crop varieties, their geographic distribution, their production quantities, and, most notably, the specific differentiations in protein attributes.

At present, despite extensive research in the manufacturing of pulse protein isolates and their functions in food products, pulse protein isolates are still underused in food processing due to their limited solubility [13,14,15]. The primary limiting factor is the poor aqueous solubility of pulse protein isolates, in which the harsh isolation conditions of preparation processes including soaking, tempering, milling, and alkaline extraction followed by isoelectric precipitation and drying can compromise protein structure and thus protein functionalities, which largely depend on solubility [10,16,17]. Therefore, it is crucial to seek appropriate strategies to effectively enhance the functional properties of pulse protein isolates. Modifications of pulse protein isolates have been demonstrated in the literature by chemical, biological, physical, or a combination of these methods as promising strategies to enhance their functionalities as alternative protein compositions to animal-derived products [9,18,19]. It is well known that protein functionalities are governed by their varied heterogeneous structures as well as the resultant physicochemical properties, i.e., amino acid sequence, secondary/ternary structures, surface potential and charge distributions, hydrophilic and hydrophobic characteristics, aggregation behaviors, etc. [9,19,20], which are all tunable for appropriate food applications [20].

In this article, firstly, the origins and general compositions of pulse crops are introduced. Then, we focused on their protein fractions, especially the relationship between protein structures and functionalities, i.e., interfacial properties, aggregation behavior, solubility, etc. Finally, various modification strategies to pulse proteins, mainly including chemical (covalent and non-covalent), physical, and biological treatment, were systematically elucidated, aiming to summarize the state-of-the-art modifications that have been attempted to boost the performance and functionalities of pulse proteins in food applications.

## 2. Materials and Methods

### 2.1. Data Sources

A thorough search was conducted to gather studies investigating strategies aimed at enhancing the functional properties of pulse proteins and broadening their applications in the food industry. The systematic review focused on published articles in the English language, excluding reviews, spanning the years 2010 to 2023, with a specific emphasis on the last three years. The databases utilized for collecting relevant articles were Web of Science (https://webofscience.clarivate.cn/wos/alldb/basic-search, accessed on 15 July 2023) and Elsevier (https://www.sciencedirect.com/, accessed on 15 July 2023). Additionally, partial worldwide data on pulse proteins and protein crystal structures were obtained from Our World in Data (https://ourworldindata.org/, accessed on 17 July 2023) and the PDB database (http://www.rcsb.org/pdb/, accessed on 17 July 2023), respectively.

### 2.2. Search Strategy

The search strategy employed the following keywords: (pulse crops OR pulse protein isolates OR pea protein OR chickpea protein OR cowpea protein OR lentils protein OR bean proteins OR faba bean protein OR mung bean protein) AND (solubility OR water holding capacity OR oil holding capacity OR emulsifying properties OR foaming properties OR gelation properties OR bioactive properties) AND (chemical modification OR complexation OR interaction OR physical modification OR biological modification) OR interaction OR [food application]. All the initial literature records were exported in full-record format. Following this, through a meticulous review of the complete texts, studies deemed irrelevant were systematically excluded. Relevance assessments were conducted by all authors, and consensus was reached through collaborative evaluation. Ultimately, a total of 473 articles that met the established criteria were retained for further in-depth analysis.

### 2.3. Software Used

Within the scope of this work, the following software applications were employed: Origin 2023 (OriginLab, Northampton, MA, USA), PowerPoint 365 (Microsoft, Bellvue, MA, USA), and Photoshop 2023 (Adobe Systems Incorporated, San Jose, CA, USA) for the generation of visual representations. Data processing was conducted using Microsoft Office 365 Excel (Microsoft, Bellvue, MA, USA).

## 3. Origins and Compositions of Pulse Crops

The term “pulse” is defined as the nutritional-dense edible legume crops that are harvested solely for dry seeds, e.g., dry peas (*Pisum sativum*), pigeon peas (*Cajanus cajan*), chickpeas (*Cicer arietinum*), cowpeas (*Vigna unguiculata*), lentils (*Lens culinaris*), common beans (*Phaseolus vulgaris*), faba beans (*Vicia faba*), bambara beans (*Vigna subterranea*), mung beans (*Phaseolus aureus*), black gram (*Phaseolus mungo*), moth beans (*Phaseolus aconitifolius*), and velvet beans (*Stizolobium* spp.) [8,12]. The genera, species, and common names of some typical pulses are systematically summarized in Table 1. According to the U.N. Food and Agriculture Organization (FAO), the annual worldwide production quantities (from 1961 to 2020) of some pulses are depicted in Figure 1A. Over the past four decades, the production of pulse crops experienced obvious upward trends; meanwhile, pulses have also become the second most consumed crops, after cereals, for human diets around the world [11,12]. Dry beans, peas, and chickpeas are the most popular strains among all pulses, and their annual production is approximately three times higher than those in 1961. Owing to the reduced moisture contents, pulses exhibit a relatively long storage life compared to fresh legumes and thus are widely cultivated all over the world [12]. Geographically, pulse crops are grown in India, North America, China, and Europe, which exhibit excellent soil and climate tolerance [11,21]. For example, beans are primarily produced in South America, North America, Asia, and Africa (Figure 1B), while most peas are grown in Asia, Europe, and North America (Figure 1C).

From the nutritional perspective, pulse crops contain carbohydrates, fibers, minerals, vitamins, and other significant bioactive substances, with >30% protein as the most noticeable attribute [12]. The macronutrient composition of several common pulses is listed in Table 2 [22,23,24,25,26,27,28,29]. While carbohydrate is typically the highest content among nutrients, protein ranks second in these pulse products. Due to their high protein content and being non-demanding in farming conditions, pulses have been staple crops in regions where meat protein is scarce. Except for chickpeas and lupins, all pulses are low in fat content (<5%, *w*/*w*) [27]. Even so, a high unsaturated fatty acid profile in pulses has been reported [22]. Combined with the high content of dietary fibers, pulses including peas have been proven to protect against cardiovascular disease and obesity [30]. In addition to macronutrients, pulses are also rich in micronutrients that are beneficial. The diverse phytochemicals, such as flavonoids, phenolics, tannins, saponins, phytates, oxalates, lectins, and enzyme inhibitors, show antibacterial, anti-tumor, anti-ulcerative, and anti-inflammatory properties in addition to suppressing cholesterol levels [31,32]. Furthermore, pulses are also abundant in vitamins and minerals, particularly iron. For example, beans are rich in vitamin K, carotene, and numerous forms of vitamin B, including folic acid, pantothenate while chickpeas are abundant in riboflavin, niacin, folate, and the precursor of vitamin A [22,33].

To raise public awareness of the nutritional and health benefits of pulses, which represent potential candidates to stimulate sustainable food development and global food supply, FAO has designated 2016 as the International Year of Pulses [31,34]. Nevertheless, protein in pulses is unarguably the primary nutritional contribution, and hence pulse proteins are elaborated on in the next section, mainly about their primary structures and composition.

**Table 1 foods-13-00006-t001:** Generic and species names and common names of pulses [22,25].

Genus	Species	Common Name
*Phaseolus*	*vulgaris*	Common bean (Kidney, navy, great northern bean)
	*lunatus*	Lima bean (Butter bean)
	*coccineus*	Runner bean (Scarlet runner)
	*acutifolius*	Tepary bean
	*dumosus*	Year bean
*Vigna*	*angularis*	Adzuki bean
	*radiata*	Mung bean (Green gram bean)
	*mungo*	Black gram bean
	*aconitifolia*	Mat bean, Moth bean
	*unguiculata*	Cowpea (Black-eyed pea)
	*subterranea*	Bambara bean (Earth pea)
*Lupinus*	*mutabilis*	Lupin
	*albus*	White lupin
	*angustifolia*	Blue lupin (Narrow-leafed lupin)
	*luteus*	Yellow lupin
*Pisum*	*sativum*	Pea
*Cicer*	*arietinum*	Chickpea
*Lens*	*culinaris*	Lentil
*Cajanus*	*cajan*	Pigeon pea (Red gram bean)
*Lablab*	*purpureus*	Lablab bean (Hyacinth bean)
*Canavalia*	*gladiate*	Sword bean
*Psophocarpus*	*tetragonolobus*	Winged bean
*Cyamopsis*	*tetragonoloba*	Guar bean
*Mucuna*	*pruriens*	Velvet bean
*Macrotyloma*	*uniflorum*	Horse gram bean

**Table 2 foods-13-00006-t002:** Macronutrient content of several common pulses (g/100 g dry matter).

	Protein	Starch	Dietary Fibre	Fat	Ash
Pea [22,25,26,27,35]	14–31	30–50	3–20	1–4	2.3–3.7
Chickpea [22,24,25,27,35]	19–27	33.6–51.7	2.9–20.75	2–7	1.8–3.5
Cowpea [24,25,27,29,35]	24–28	33.1–63.6	10.06–34	1.26–2.22	2.9–4.4
Pigeon pea [27,35]	19.3–22.4	NR	6.4–7.25	2.74	0.04–2.13
Lentil [22,25,27,35]	23–31	37–59	7–30.5	1–3	2.1–3.2
Lupin [22,25]	32–44	1–9	14–55	5–15	2.6–3.9
Faba bean [22,26,28,35]	19–39	27–50	25–29.6	1.53–3.2	1.14–7.1
Mung bean [23,24,35]	14.6–32.6	29–58	3.8–6.15	0.17–7	0.17–5.87
Common bean [25]	17–27	31–43	18–30	1–5	3.2–5.2

## 4. Composition and Structure of Pulse Protein Isolates

### 4.1. Amino Acid Composition

The nutritional properties and functional characteristics of pulse protein isolates are determined by their amino acid compositions and sequences (primary structure) as well as the derived higher-level structures, i.e., secondary, tertiary, and quaternary, during their folding and complexation. On this basis, the additional advanced structures are assembled through dynamic bonding such as hydrogen bonds, hydrophobic contacts, electrostatic interaction, and disulfide bonds [36]. The amino acid (AA) composition of several selected pulse proteins is given in Table 3 [9,11,25,27,37,38,39,40]. It is notable that the contents of essential amino acids, including lysine, leucine, aspartic acid, glutamic acid, and arginine, are relatively high in pulse proteins. Particularly, lysine, a well-known limiting essential AA in cereals, is abundant in pulse proteins; for example, the lysine content is about 7.7 g per 100 g in chickpea and pea proteins [37,38]. However, according to the sequences, pulse proteins are deficient in two essential AAs, methionine and tryptophan. Therefore, it is viable to complement pulse with tryptophan-rich proteins such as canola protein to offer a complementary essential AA composition [11,40]. Variations in AA profiles of different pulse proteins are caused by their species, growth environments, and differences in measurement methods [12,27,37]. For example, aspartic acid (Asp) and glutamic acid are the most abundant in pea, chickpea, lentil, mung bean, lupin, and cowpea proteins, while the content of Asp is relatively low in faba beans. According to Tang et al., the protein isolates from lentils, peas, and pigeon peas have a higher content of essential AAs (Leu, Lys, Ile, Met, Phe, Val, Thr, and Try) than other pulse proteins [27]. Ge et al. found that panda bean (*Vigna umbellate (Thunb.) Ohwi et Ohashi*) protein isolate presents an excellent amino-acid composition and protein efficiency ratio [41]. Additionally, the different ratios of hydrophilic and hydrophobic AAs greatly affect the protein secondary structure, spatial structure, and functional properties of pulse protein isolates [21,37,42].

### 4.2. Protein Fractions and Structures

Based on solubility in water, saline, dilute acid or alkali, and alcohol, pulse proteins are empirically divided into four primary fractions known as albumin, globulin, glutelin, and prolamin, respectively [24,43]. The reported range of primary protein compositions is given in Table 4 [23,25,39,44,45,46,47]. Pulse protein isolates are predominantly constituted by globulin and albumin at approximately 50–80% and 10–20% of total storage proteins, respectively, where glutelin (10%) and prolamin (less than 5%) are minor constituents. These subunit compositions vary considerably in structures and functions [42,48], which are presented in greater detail below.

Albumins are small, compact globular units, ranging in molecular weight from 5 to 80 kDa, and typically composed of two polypeptide chains. They have a three-dimensional structure that is abundant in α-helices and a well-preserved skeleton composed of eight cysteine residues [9]. Protease inhibitors, lectins, amylase inhibitors, and enzymatic proteins are common in pulses belonging to albumins [45,49]. As shown in Table 4, pea and faba beans contain higher albumins than other pulses. Nutritionally, albumins provide a good supply of essential amino acids (tryptophan, lysine, and threonine) and a higher percentage of sulfur-containing amino acids (cysteine and methionine) than globulins [49,50]. However, Ghumman et al. found that albumins presented lower in vitro digestibility than globulins when they compared the functional properties of different subunit fractions from pulse crops [48]. Albumins also presented better foaming properties in pulses due to their excellent aqueous solubility [48].

Globulins are the dominant protein components found in pulses and can be further classified into main legumin (11 S) and vicilin (7 S) proteins based on their sedimentation coefficients (S = Svedberg Unit), with minor levels of a third type known as convicilin [9,21].

Legumin is a hexameric protein with a stiff conformation and considerable quaternary structure, with a molecular mass of 300 to 400 kDa [9,51]. It is composed of six subunit pairs (each around 60 kDa), which interact noncovalently, and is further assembled through two trimeric intermediates (Figure 2A). Each legumin subunit is constructed from two parts: a heavy acidic α-chain of ~40 kDa and a light basic β-chain of ~20 kDa, connected by a disulfide bond. This structure can dissociate when reacted with reducing agents [9,11]. According to the composition of hydrophilic and hydrophobic amino acids, the α-chain is predominately glutamic acid and has leucine as the N-terminal amino group, while the β-chain has more alanine, valine, and leucine and has a glycine terminal. As a result, the hydrophilic acidic subunits are mostly exposed in aqueous solutions, while the basic subunits are embedded in the inner hydrophobic cavity [21,37,52].

Vicilin is an oligomeric protein with a trimeric structure that ranges in molecular mass from 150 to 190 kDa. Each monomer is between 50 and 70 kDa and consists of three subunits (α, β, and γ), which are linked together primarily by electrostatic forces and noncovalent hydrophobic interactions [53]. Due to the deficiency of sulfur-containing AAs, no disulfide bond is present between vicilin protein molecules. Nevertheless, vicilin includes significant amounts of arginine, lysine, aspartic acid, and glutamic acid [21,37]. As an example, demonstrated in Figure 2B, a pea vicilin monomer can be divided along a pseudo-dyad axis into two homologous components that share a core region and extended arms to form N- and C-terminal domains. That central region is composed of β-barrels, while its extended arms are comprised of α-helices. In addition, each monomer has a core region that is established by α-helices and β-barrels extending from the core and combining with the neighboring monomers to form a trimeric structure [53]. Despite certain similarities between pulse vicilin, there is a large variance in mass, surface charge, and glycosylation of proteins. For example, vicilin proteins from lentils and cowpeas are glycosylated, while faba bean lacks carbohydrates in its structure [43,54]. Shrestha et al. identified the protein fraction composition of lentils and yellow peas based on molecular weight estimates from sodium dodecyl sulfate–polyacrylamide gel electrophoresis analysis [55]. They found that soluble proteins from them were identified as legume-like and vicilin-like, whereas vicilin-like proteins predominated in mung bean.

Convicilin, a constituent of globulin, has been found as the third minor storage protein in pulses, in addition to legumin and vicilin [56]. A convicilin molecule is about 70 kDa and is often found as a trimer of about 210 kDa (or 290 kDa including an N-terminal extension) comprised of three convicilin molecules or as heteromeric trimers of convicilin and vicilin (Figure 2C). The structure and composition of convicilin are distinct from that of both legumin and vicilin, however; it contains sulfur-containing AAs, unlike vicilin, and a high charge density in the N-terminal extension [9,53,57].

Glutelin and prolamin are also present in trace amounts in pulses. Glutelin contains considerable quantities of methionine and cysteine, unlike globulin. The abundance of sulfur-containing AAs facilitates the formation of disulfide bonds between protein molecules, which promotes aggregation [58]. It was reported that the high glutelin content in grains, such as rice, is associated with low aqueous solubility [59]. Notably, chickpeas have a relatively high glutelin content (19–25%) among pulse crops (Table 4) and, therefore, have been reported for relatively low protein solubility [14,60]. On the other hand, prolamin is an alcohol-soluble protein with a high proportion of proline and glutamine, like most cereal proteins, comprising a minor portion (less than 5%) of the total proteins in pulses [61]. Therefore, it is believed that pulses with high glutelin and prolamin concentrations have better protein quality [25,48].

Despite the similarities in protein composition among different pulses, the functionalities of pulse protein isolates are greatly affected by these slight alterations from variety and growth environments [12,27,37]. Albumins promote the foaming characteristics of pulse protein isolates, whereas globulin has the opposite effect [48]. Other functional features, such as emulsification and gelation, are also significantly affected by overall protein structure. For instance, some research found that yellow pea vicilin, which has higher water solubility and surface hydrophobicity than legumin, leads to enhanced emulsifying properties [42,62]. Additionally, it was asserted that yellow peas with high globulin contents have better protein extractability due to their excellent solubility. Likewise, pea vicilin demonstrates an appropriate capacity for heat gelation, unlike legumin [42,62]. Therefore, pulse protein isolates with a high vicilin/legumin proportion can be preferable for their food application as functional ingredients. In summary, the various structures of pulse proteins determine their functional properties, which further influence their practical applicability and competitiveness in the market. In the next section, the functionalities and current food applications are discussed.

## 5. Functionality and Food Application of Pulse Protein Isolates

The functional properties of pulse protein isolates determine their eventual use in the food industry and play a significant role in food nutrition, sensory, texture, and organoleptic qualities. Among these properties, solubility, water/oil-holding capacity, emulsifying properties, foaming, gelation, and bioactivity are the most crucial functional qualities and are receiving wide attention [10,21]. In many reports, these characteristics differ greatly depending on the source and protein structure, as mentioned before, as well as the processing conditions (pH, temperature, and ionic strength) [27,37]. An overview of these functionalities is provided below.

### 5.1. Solubility

Solubility is one of the most important protein properties affecting its bioavailability and other related functionalities, such as interfacial characteristics, digestibility, and gelling properties [11]. The ratio of hydrophilic to hydrophobic residues and their arrangement in AA sequences determine how soluble the protein molecule is in aqueous media. Protein solubility is hampered by the formation of aggregates, which are brought on by hydrophobic interactions between protein molecules caused by hydrophobic surface patches [7,30]. In addition, pH, temperature, type, and strength of the salt ions, as well as other factors in the solution environment, all have a significant impact on the solubility of pulse proteins [33,43]. In terms of pH, proteins are least soluble at their isoelectric point due to a zero net surface charge, which causes protein molecules to aggregate into bigger structures. On the other hand, when the pH values are higher or lower than the protein’s isoelectric point, proteins exert a negative or positive net charge on the solution, and the electrostatic repulsion between charged molecules promotes the solubility of the proteins [42]. V-shaped solubility characteristics against pH, with better solubility under extremely acidic (below pH 3) or alkaline (above pH 9) environments and lowest solubility at the isoelectric point (pH 4–5), have been reported for lentil, green mung bean, pigeon pea, cowpea, pea, and chickpea protein isolates (Figure 3A) [27]. The surface charge (zeta potential) and electrostatic repulsion of pulsed proteins can be affected by ions in solution, which then impact their solubility. It has been noted that while sulfate, hydrogen phosphate, ammonium, and potassium salts promote ion–water interactions, thiocyanate, perchlorate, barium, and calcium salts encourage protein–water interactions and organize the hydrated layer surrounding the protein to stimulate solubility [12,43].

### 5.2. Water/Oil Holding Capacity

The terms “water holding capacity” (WHC) and “oil holding capacity” (OHC) describe how much water and oil, respectively, can be absorbed per gram of pulse protein. As with solubility, the WHC and OHC of proteins also are determined by the ratio of hydrophilic to hydrophobic amino acids on protein particles’ surfaces [11,33,44]. These two characteristics are crucial when evaluating the quality, texture, and mouth feel of pulse protein products. For example, pea protein isolates (PPIs) with high WHCs were employed to stabilize the gel structure of a dough [63], in which the absorbed water is used to prevent flour from dissolving. Additionally, pulse protein isolates with high OHCs, such as pea, lentil, and faba bean, are often applied in meat/fat analogs and bakery products [64,65]. In general, protein isolates from most pulse crops present higher WHCs and OHCs than those of flour [11,42].

### 5.3. Emulsifying and Foaming Properties

Pulse proteins have both emulsifying and foaming properties, which are both extensively used in food. These two characteristics, like WHC and OHC, are influenced by proteins’ amphiphilic nature [11,12]. An emulsion is a mixture of two immiscible liquid phases, typically water and oil, in which one liquid is distributed inside the continuous phase of the other. Due to their different densities and immiscibility, the two phases’ interface is thermodynamically unstable. The applications of pulse proteins in emulsion-based foods like milk analogs, batters, cakes, soups, and mayonnaise require their capability of forming or retaining a stable oil/water interface. By creating an interfacial film around oil phases diffused in an aqueous system, pulse protein isolates could function as emulsifiers, preventing structural changes like coalescence, creaming, sedimentation, or flocculation [39,62]. Two indexes are frequently used to assess the emulsifying capabilities of pulse protein isolates: emulsifying activity (EA) and emulsifying stability (ES). EA quantifies how much oil can be emulsified per unit of protein, while ES quantifies the emulsion’s capacity to withstand structural changes over a predetermined period. Emulsifying qualities of the protein isolates in various pulses and their varieties vary greatly. Tang et al. [27] compared the EA and ES of pulse protein isolates from lentils, green mung beans, pigeon peas, cowpeas, peas, and chickpeas. Pea proteins were found to possess the best EA and ES of 0.76 and 0.62 cm/cm (heating for 30 min), respectively. Ground peas, kidney beans, cowpeas, lentils, and horse gram protein isolates had EAIs and ESIs ranging from 4.7 to 26.6 cm^2^/g and 7.2 to 95.4 min, respectively. [48]. Electrostatic charge repulsion (which is dependent on the surface charge distribution and pH) and the continuous phase viscosity both have an impact on emulsifying stability. A study on mung bean protein isolates revealed that the pulse proteins had a higher ES in acidic environments (pH 3) and a higher EA in alkaline conditions (pH 10), with the worst emulsification properties at the isoelectric point [66].

Foaming is crucial in some specific food applications, such as milk tea, whipped toppings, mousses, chiffon cakes, ice cream mixes, etc. [67]. Foam is a dispersion of gas bubbles formed when air bubbles are trapped by thin liquid layers [68]. Foam generation depends on the interfacial tension between two immiscible phases (aqueous and air), just as emulsions, and requires an energy input (sparging, whipping, or shaking) [14,60]. Foams are thermodynamically unstable because of the large free energy present at the gas–liquid interface, which causes them to agglomerate and become disproportionate, thereby decreasing the interfacial area. Due to their capacity to lower surface tension from the amphiphilic properties and create sturdy interfacial membranes through protein–protein interplay, pulse protein isolates can stabilize the air/water interfaces of foams. The foaming properties of pulse protein isolates are evaluated by their foaming capacity (FC) and foam stability (FS), where FC is the ratio of the volume of the whipped foam of the protein solution to the solution volume, and FS is the amount of time needed for the foam to lose a specific amount of volume [27,67]. The source of protein, environmental factors (like temperature and pH), and whipping strength all affect foaming performance. Different pulse proteins (beans, peas, and chickpeas) exhibited greater foaming in the acidic and alkaline pH ranges while exhibiting lower values at pH levels near the isoelectric point [21,69]. Tang et al. [27] reported the FCs and FSs of protein isolates from different pulses (Figure 3B) and found that mung bean protein isolates demonstrated a higher original FC. Cowpea and chickpea protein isolates had relatively poor foaming qualities, while lentil, mung bean, pea, and pigeon bean protein isolates showed excellent FSs (above 0.8 mL/mL after 90 min resting) [27]. Toews and Wang also reported that chickpea protein isolates had a 201–228 percent foaming capacity, but these percentages were noticeably lower than those for other pulses [70]. Ge et al. reported that panda bean protein isolates presented superior emulsifying and foaming abilities, compared to soy protein isolates and pea protein isolates [41]. As previously expounded upon, pulse proteins exhibit discernible variations in their foaming properties, attributable to disparities in protein composition and structural attributes. However, a comprehensive exploration elucidating the specific protein structures and compositions that promote improved foaming capacity and stability is currently lacking, which is worthy of in-depth research.

### 5.4. Gelation Properties

Gel is a three-dimensional spatial network structure formed by the interaction between molecules and polysaccharides, and protein combinations are the most typical gelling composites in food products. Gel-like food products retain their unique structure and resist flow under force, especially heating and then cooling [60,67]. Gelation controls morphology, texture, and viscoelasticity, which affect foods’ general rheological and taste attributes. In viscous products like mousse, soup, gels, curds, and meat substitutes, gelation is a crucial functional characteristic of pulse protein isolates. The interaction of heat-induced denatured protein molecules to form a three-dimensional spatial structure that encloses water, oil, and other food matrices is the primary cause of the gelation of pulse proteins under temperature changes [71]. The protein content needed to produce a stable gel from a liquid is defined as the least gelling concentration (LGC), and it is used to measure the gelation properties of pulse proteins. Therefore, proteins with lower LGC values have a better capacity to form stable gel structures. Protein concentration, pH, ionic strength, amino acid ratio, and interactions with other elements are just a few of the variables that affect the thickening process of protein gels. The LGC values of protein isolates from different pulses were measured in the range of 80 g/L (pigeon bean) to 160 g/L (mung bean) [27]. Previous work found that proteins from cowpeas, chickpeas, and pigeon beans presented relatively better gelation properties due to processing more ordered secondary structures such as α-helices, β-sheets, and β-turns than the others [27]. For chickpea protein isolates, the effects of pH, ionic strength, and ionic species were examined. The LGC value needed to form a gel at pH 3.0 was higher (180 g/L) compared to that in neutral environments (140 g/L) and adding 0.1 M NaCl significantly decreased the LGC value. Additionally, the findings revealed that the gel strength for the samples containing CaCl2 was greater than that for the samples containing NaCl at pH 3.0, meaning the type of cation has an impact on the gelation process [37,72].

### 5.5. Bioactive Properties

Pulse protein isolates are widely used as food ingredients mainly due to their macronutrient supplementation and physicochemical functional properties. However, due to the intensive development of the protein’s biological activity in recent years, it has caught the increasing attention of researchers. Pulse proteins are considered to have antimicrobial properties as well as the ability to reduce the risk of certain diseases, such as type 2 diabetes, metabolic syndrome, and obesity [73]. Pulse proteins’ ability to interact with elements of bacterial, fungal, or viral cells is what is thought to be responsible for their antimicrobial activity, such as the binding of lectins with hyphae [73]. Abdel-Shafi et al. proved that the 7S and 11S globulins from cowpea, employed in minced meats, were inhibitory to several foodborne spoilage and pathogenic bacteria with a minimum inhibitory concentration (MIC) of 10 to 200 µg/mL [74]. In addition, pulse lectins from lentils and lablab beans exhibit activities against some viruses, such as severe acute respiratory syndrome coronavirus-2 and influenza virus [75,76]. Meanwhile, lectins, hemagglutinins, and enzyme inhibitors in pulse protein isolates have been demonstrated to help in reducing serum glucose levels and alleviating obesity [73]. By decreasing peroxisome proliferator-activated receptors, decreasing adiposity, favorably influencing adipokines, and enhancing short-chain fatty acid-producing microorganisms in the intestine, chickpea protein isolates may prevent adipogenesis and raise glucose transporter-4 levels while decreasing insulin sensitivity [31]. Additionally, it has been discovered that replacing animal proteins with pulse proteins lowers levels of apolipoprotein, non-high-density lipoprotein, and low-density lipoprotein cholesterol, which are linked to cardiac diseases [21,77].

Indeed, additional research has revealed that the pulse protein peptides produced by protein degradation have increased biological activity, especially in antihypertensive, antioxidant, anticancer, and antidiabetic properties [78]. According to Daskaya-Dikmen et al. [79], bioactive peptides from pulse crops inhibited the angiotensin-converting enzyme (ACE) and significantly changed the substrate’s C-terminal peptide sequence (Figure 4A). The growth of colorectal cancer cells in vitro was reported to be inhibited by bioactive peptides from common beans through the loss of mitochondrial membrane potential, depolarization of the mitochondrial membrane, and increased generation of intracellular reactive oxygen species (Figure 4B) [78]. Moreover, bioactive peptides derived from black beans showed inhibitory potential on DPP-IV, which could inactivate incretin hormones leading to diabetes (Figure 4C) [80]. Owing to these biological activities, an increased consumption of pulse proteins in the diet could be considered an effective promotion of health benefits.

### 5.6. Food Application

Traditionally, pulse crops, as a staple food along with cereals in many parts of the world, were mainly consumed with simple cooking such as soaking and boiling [38]. Nowadays, with the advances in food processing technology and the requirement for precision health, pulse protein isolates are separated from grains and then used solely as raw materials or food additives in the formulated products [12,40]. Nadeeshani et al. reviewed the utilization of pulse protein in food and industrial applications [38]. As given in Figure 5, various pulse protein isolates are employed in different types of food processing, such as animal-source food alternatives, bakery goods, snack foods, and nutritional products [44,45,62,63,64,67]. For instance, Schoute et al. [81] discovered that substituting 20% to 40% wheat flour for chickpea protein flour can effectively lower the production of acrylamide during biscuit preparation, as well as preserve the color and texture of the biscuits. In addition, pea peel protein was recognized as a value-added food ingredient to produce healthy snack crackers and dry soup [82]. Furthermore, research has explored the use of yellow pea and red lentil flours to create high-quality, nutritionally dense expanded cellular snacks [83].

Among these applications, choosing the appropriate pulse protein isolates with processing technology in processing animal protein alternative foods, such as meat, egg, and milk analogs, is the most discussed topic currently [22,62]. Due to the low resource consumption and high nutritional benefits, replacing animal proteins with plant proteins is considered environmentally friendly and beneficial for sustainable agriculture [77]. For example, pea and mung bean protein isolates were used instead of milk to produce plant-based yogurt that possessed a good flavor profile and taste quality [84]. Ramos-Diaz et al. investigated the application of faba bean protein isolates in meat-free alternatives to minced meat [85]. They found that plant-based substitutes for minced meat presented comparable or higher mechanical properties than beef minced meat, which confirmed the potential utilization of pulse protein isolates in meat analogs. In recent years, two well-known commercialized meat substitute brands on the market, Beyond Meat, and Impossible Foods, have gained a tremendous growth in popularity after launching whole plant-based burgers in 2016 [38]. In food products like meat burgers, sausages, fish balls, chili, pizza toppings, and meat sauces, pulse proteins have been frequently added to replace real meat. These same techniques are also used in the development of egg substitutes by adding mung bean protein isolates [86]. Moreover, these pulse-based products contain higher protein and vitamin amounts as well as additional nutritional advantages like higher dietary fiber and lower sodium, cholesterol, and calories over animal-based products [2,64,77].

## 6. Modification Strategies of Pulse Protein Isolates

Pulse protein isolates have shown great potential in various food applications, as demonstrated in many laboratory studies [21,37]. However, commercial pulse protein isolates are produced under harsh conditions, which usually cause protein denaturation and poor solubility, thus creating negative impacts on the performance of other functional properties in food products [9,42]. A multitude of studies suggests that the functional profile of pulse protein isolates can benefit from protein structural modification (chemical, physical, and biological methods, or others) [87,88]. Therefore, it is essential to develop technically and economically sustainable approaches to improve pulse proteins’ functional properties to increase their use in food processing. An overview of protein modification techniques and their effects on pulse proteins’ functionalities is given in the following sections.

### 6.1. Chemical Covalent Modifications

Chemical covalent modification is an unambiguous strategy for precisely altering pulse protein structure to improve functional properties. Typically, chemical covalent modifications produce tailorable functionalities by selectively incorporating functional groups on protein side chains through reactive residues of interest. Currently, pulse protein isolates have been reported to undergo various chemical covalent modifications, mainly including acylation, amidation, esterification, glycation, and phosphorylation. As seen in Figure 6, Zha et al. described the simplified reaction process of these chemical covalent modification methods [9].

Protein acylation is the process of adding acyl groups to protein molecules, and acetylation and succinylation are the two main forms that have been successfully performed on pulse protein isolates. Shah et al. performed a hydrophobic modification of pea proteins by using succinic anhydride, octenyl succinic anhydride, and dodecyl succinic anhydride [87]. Modified pea proteins exhibited better functional properties and performance as additives in an eggless cake formulation. Charoensuk et al. indicated that succinylation at low succinic anhydride addition altered mung bean protein charge and significantly improved emulsifying properties [88]. The process of glycosylation entails the affixing of carbohydrate moieties to lysine residues or the N-terminus of a protein, which is usually accompanied by the Maillard reaction. Caballero and Davidov-Pardo suggested that Maillard conjugation could improve the emulsification properties of pea protein isolates [89]. Additionally, Zhao et al. utilized the Maillard reaction to enhance the functionality of pea protein isolate by covalently linking it with xylo-oligosaccharides [90]. Phosphorylation introduces a phosphoryl group (PO_3_) functional group to a specific reactive amino acid residue (-NH, -OH, or -SH) on a protein molecule through a covalent bond, and this functionalization enhances the hydrophilicity of proteins by increasing their negative surface charges. Liu et al. reported the phosphorylation of pea protein isolates with improved solubility, emulsifying property, emulsifying stability, foaming property, and oil absorption capacity, thus expanding the application of peas in the food industry, such as fat mimics [91]. However, chemical covalent modification approaches are still limited in scaled-up food production due to modification costs including high consumption of chemical reagents and long reaction time, safety risks, and clean label requirements.

### 6.2. Non-Covalent Complexation Modifications

Non-covalent dynamic bonds form through intermolecular forces or interactions with substances, such as protein–protein, protein–polysaccharide, and protein–polyphenol interactions, resulting in protein conformation changes as well as the formation of protein complexation [92,93,94]. Current research highlights the potential of combining pulse proteins with other edible components, such as polymers or small molecules, in order to construct multicomponent molecular complexes, thus improving the quality and nutritional value of food products [92,94,95,96].

Pulse proteins contain hydrophobic groups that spontaneously form hydrophobic cavities in aqueous solutions, allowing non-covalent interactions with hydrophobic small molecules like epigallocatechin-3-gallate (EGCG), rutin, quercetin, chlorogenic acid, and resveratrol [93,94,95,97]. In a study by Hao et al., the presence of polyphenols improves the foaming, emulsification, and in vitro digestibility of pea protein isolates [93]. Similarly, Han et al. observed enhanced interfacial properties in PPI–EGCG complexes compared to pea protein alone [95]. In addition to polyphenolic compounds, specific hydrophilic small molecules, such as arginine [98], have been shown to enhance protein functionality. Cao et al. [98] found that adding 0.2% arginine altered the PPI structure, resulting in improved emulsification and a 20% increase in protein solubility. Moreover, the interaction between pulse proteins and edible polymers can also significantly enhance functional properties, such as solubility and emulsification [92,99,100]. For instance, carboxymethylcellulose increased mung bean protein solubility from 1.69% to 43.62% due to stronger hydrogen bonds between protein/polysaccharide complexes with water [99]. Interestingly, protein–protein interactions have been highlighted for pulse protein modification in recent research [13,15,17,101]. With pH-shifting from pH 12 to pH 7, the fabricated pea protein–rice complex demonstrated improved solubility and enhanced nutritional values [13]. In another study, when PPI was in a mixture with whey protein isolates, an increase in the nutritional and functional properties of PPI was also observed by Kristensen et al. [101]. Alrosan et al. improved lentil protein solubility by combining it with quinoa proteins at pH 12 [17]. Also, Teng et al. designed a binary whole pulse protein complex when co-assembling pea protein with chickpea protein [15]. The novel binary protein presented superior solubility (50% higher than chickpea protein alone) due to the interplay between unfolded chickpea protein and pea protein during pH shifting, which enabled their resistance to acid-induced structural over-folding.

In contrast to chemical covalent modification, non-covalent complexation strategies are typically conducted under mild reaction conditions, offering simplicity and ease of operation. Therefore, non-covalent complexation modification to enhance pulse protein functionality has gained significant attention from researchers and holds promise for practical applications in the food industry.

### 6.3. Physical Modifications

Novel physical processing technologies have emerged as alternatives to traditional heat or chemical modifications for improving pulse protein functionalities, often bearing the label of ‘clean’ and ‘additive-free’ [102,103]. Generally, physical modifications can be categorized into thermal (such as microwave heating, radio frequency heating, ohmic heating, and infrared heating) and non-thermal (including ultrasonication, cold plasma, pulsed electric fields, and high hydrostatic pressure) processes [16,104]. Non-thermal modification is garnering substantial attention due to its innovative attributes: it minimizes damage to nutritional and sensory properties with advantages in cleanness, sustainability, and low energy consumption [105]. The schematic of the effect of physical modification on pulse protein conformation is shown in Figure 7 [16].

(1)Ultrasound induces cavitation and microstreaming currents, generating high temperatures and pressures for pulse protein modification, altering its spatial structure to enhance functionality. This includes heating and localized hydrodynamic shearing of protein molecules in a solution [18]. Many studies have demonstrated that when pulse protein, such as pea [106,107] and chickpea protein [59,108], was subjected to ultrasound treatment, it often leads to improved solubility and superior interfacial properties at both oil–water and gas–liquid interfaces.(2)Cold plasma constitutes the fourth state of matter, composed mainly of charged ions, free radicals, and electrons, which can induce protein modifications such as oxidation, cleavage, and polymerization, thus impacting protein structure [18]. Additionally, cold plasma modification can cause carbonylation and the cleaving of protein backbone peptide bonds. Bu et al. investigated the effect of cold plasma treatment on the structure and functionality of pea protein [109]. It was found that cold plasma modification increased the surface hydrophobicity of the protein and resulted in the formation of soluble aggregates through disulfide linkages. Altered protein secondary structures contribute to significant enhancements in gelation and emulsification properties [109].(3)A pulsed electric field (PEF) involves applying a strong electric field (>0.1 kV/cm) between two electrodes to a sample for a duration from milliseconds to nanoseconds [16]. Structural changes in pulse-treated proteins are driven by the response of charged chemical groups attempting to realign with the electric field through electrochemical reactions and polarization effects [110]. Numerous studies show that these external electrical fields can significantly alter both secondary and tertiary protein structures [111,112,113]. Chen et al. investigated the impact of a PEF on pea proteins and their binding capacity to EGCG through computer-based computational simulations [111]. As shown in Figure 8, PEF treatment (10 kV/cm) enhanced the binding affinity of pea protein isolates with EGCG, increasing the binding constant by 2.35 times and binding sites from 4 to 10 [111]. The number of amino acid residues involved in hydrophobic interactions in PEF-treated pea protein increased from 5 to 13.

(4)High pressure modifies pulse protein through compression, disrupting noncovalent interactions, forming new non/semi-covalent bonds, and affecting factors like hydrogen bonds, electrostatic interactions, hydrophobic interactions, and semi-covalent bonds like disulfide bonds, ultimately shaping pulse protein conformation [18]. Hall et al. explored the effect of high-pressure modification on the structure and functionality of lentil, pea, and faba bean proteins, and 4 min pressure treatment (600 MPa, 5 °C) resulted in superior solubility, water-holding capacity, emulsifying, and foaming properties of pulse proteins [114]. Similarly, cowpea protein treated with high hydrostatic pressure (400 or 600 MPa) exhibited better gelation properties [115].

Although physical modification holds the potential for promoting the functional properties of proteins, its practical application in pulse protein processing remains constrained. This limitation stems from the cost and complexity of physical field equipment at a large scale for food production. Hence, a paramount priority is conducting comprehensive research into the development of cost-effective, practical, and efficiently manufacturable physical field equipment to address the bottleneck for industrial scale-up and commercialization.

### 6.4. Biological Modifications

Biological modifications of pulse proteins involve the alteration of their primary structure, primarily targeting amino acid residues and polypeptide chains using biological agents, including proteolytic enzymes, non-proteolytic enzymes, and microorganisms [9,18]. Biological methods are preferred in food product development for their gentle reaction conditions, substrate specificity, and selectivity, which reduce the likelihood of adverse reactions. As a result, biological approaches have gained increasing attention for modifying pulse proteins.

One common method is protease hydrolysis, which involves the cleavage of specific peptide bonds with the addition of water molecules, leading to a reduction in molecular weight (Mw). Protease hydrolysis has been demonstrated to enhance the functional properties of pulse proteins, resulting in a more flexible and loosely structured protein. Various proteolytic enzymes (e.g., papain, trypsin, alcalase, and neutrase) have been used in the attempted hydrolysis [116,117]. Treated pulse proteins showed significantly improved solubility, foaming, and emulsifying properties. These enhancements are attributed to the increased flexibility when the protein molecular weight decreases, which allows the molecules to have superior adsorption capabilities at oil–water or gas–liquid interfaces, leading to an improved interface stability [118]. Additionally, Liu et al. evaluated the antioxidant activity of mung bean protease hydrolysate through ABTS, hydroxyl scavenging, and Fe^2+^-chelating activity analysis and found that mung bean alcalase hydrolysate exhibited the highest antioxidant activity, making it a promising application in the food industry [119]. Wang et al. prepared a chickpea protein hydrolysate by proteolytic hydrolysis and found that this hydrolysate had excellent cryoprotective effects on frozen surimi [120]. The chickpea protein hydrolysate alone (4%, *w*/*w*) exhibited comparable cryoprotective performance to that of the commercial formulation (4% sucrose and 4% sorbitol).

Microbial fermentation, a traditional modification method in food production, is used to boost the nutritional value of protein-based foods and eliminate earthy off-flavors associated with pulse crops [9,18]. This process also leverages protease production by microorganisms, leading to protein hydrolysis into smaller amino acids and peptides. For instance, Arteaga et al. [116] employed lactic fermentation with *Lactobacillus plantarum* to treat pea protein isolates, resulting in reduced characteristic off-flavors and immunogenicity. And *Lactobacillus plantarum* was also employed to ferment lentil flour, improving the overall health potential of lentil protein, including bioaccessibility and antioxidant activity [121]. 

Additionally, non-proteolytic enzymes, such as transglutaminase (TGase), can catalyze the intra- or intermolecular cross-linking of proteins through forming ε-(γ-glutamyl) lysine (ε-(γ-Glu) Lys) isopeptide bonds, significantly enhancing the protein gelation properties [122]. Sun and Arntfield used TGase to lower the minimum gelation concentration of pea protein from 5.5% to 3% (*w*/*v*) [123]. The resulting pea protein gel exhibited increased gel strength and elasticity, confirmed by the increased magnitudes of both G′ and G″ modulus. Zhan et al. reported that more pea protein was retained inside the network under TGase treatment, leading to a denser internal structure for emulsion gel [124]. Moreover, glutaminase, another non-proteolytic enzyme, was used for protein deamidation, converting amide groups into carboxylate groups [18]. This increases the charge density of the protein molecule, which further reduces the isoelectric point of protein and exposes hydrophobic regions in the protein structure. As a result, glutaminase-treated proteins have been reported to exhibit increased solubility and improved sensory properties [125].

## 7. Conclusions and Future Research Perspectives

Although many approaches have been developed and attempted to modify pulse protein isolates to improve their functionality in food products, a deep understanding of the composition and structure of pulse proteins is the key to further maximizing their utilization, as well as finding more approachable, scalable, and economic methods to increase their utilization. Till now, the interfacial behaviors, gelation properties, and hydration effects of most pulse proteins, including peas, beans, chickpeas, and lentils, have been widely reported. Meanwhile, the obtained protein powders, hydrogels, or dispersions are playing an increasingly vital role in food formulations. It is also noteworthy that though the clean-label nature and health benefits of pulse proteins are obvious advantages over many other plant/animal resources, the functionalities of pulse proteins are still less competitive. For example, the poor solubility of chickpea protein hinders its use as an aqueous ingredient, and the interfacial stability of pea protein is still worth further improvement compared with that of soybean. Thus, to enable more practical uses of pulse proteins in foods, modification is necessary. And the diversified origins and protein structures of pulse proteins also lead to the distinguished physicochemical properties among different pulse protein isolates. Developing effective and efficient screening mechanisms and workflows is essential considering the diversity of pulse protein products. To address these challenges, research in matching a specific protein candidate for certain food applications can be critical in filling the gap, for example, i.e., chickpea protein for cryoprotectants, lenticel for binders, pea protein for interfacial stabilizers, etc. Moreover, novel and green modification strategies for pulse proteins are still highly desirable, in which non-covalent multicomponent-based complexation, physical field-assisted modification, and enzymatic modifications hold promise. These three aspects of modification strategies, along with suitable applications, of pulse proteins merit further investigation and are worthy of a more in-depth review discussion.

## Figures and Tables

**Figure 1 foods-13-00006-f001:**
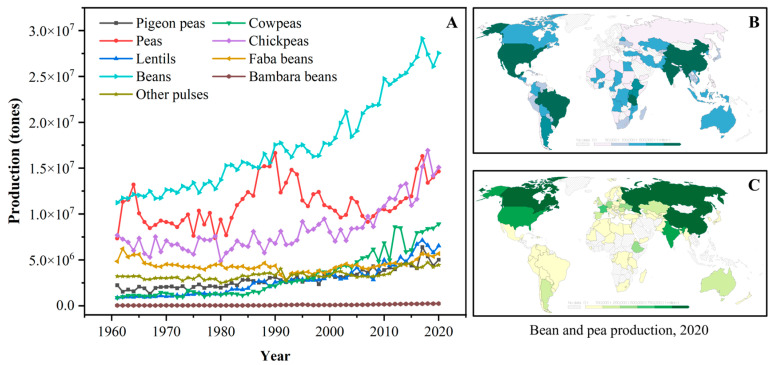
(**A**): The annual production quantity of pulse crops worldwide from 1961 to 2020 (FAO, 2020); (**B**): The distribution of bean production worldwide in 2020; (**C**): The distribution of pea production worldwide in 2020 (data obtained from Our World in Data, https://ourworldindata.org/, accessed on 17 July 2023).

**Figure 2 foods-13-00006-f002:**
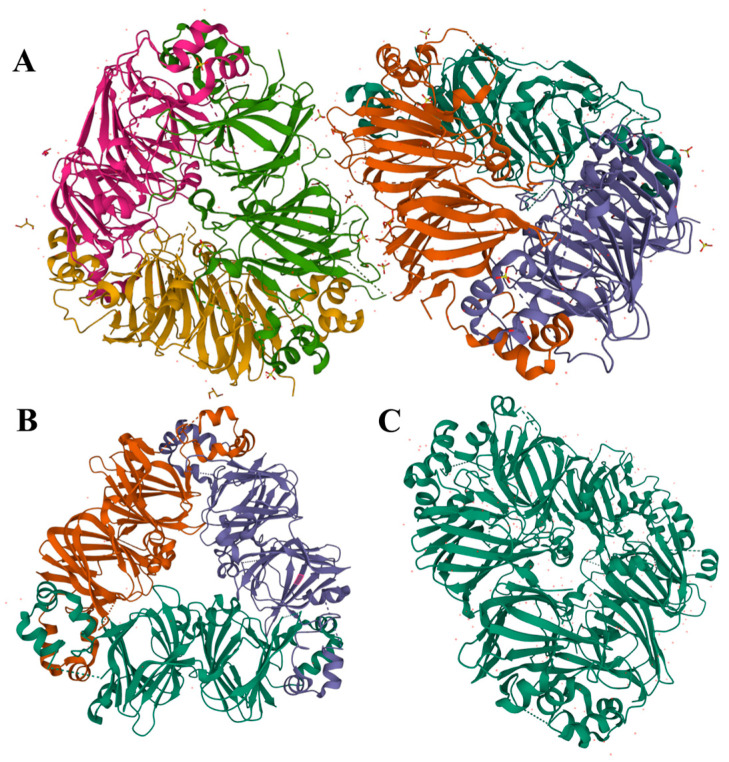
Diagram of the crystal structure of pea globulin from the PDB database (http://www.rcsb.org/pdb, accessed on 17 July 2023); (**A**): legumin (PDB entry: 3KSC); (**B**): vicilin (PDB entry: 7U1I); (**C**): convicilin (PDB entry: 7U1J).

**Figure 3 foods-13-00006-f003:**
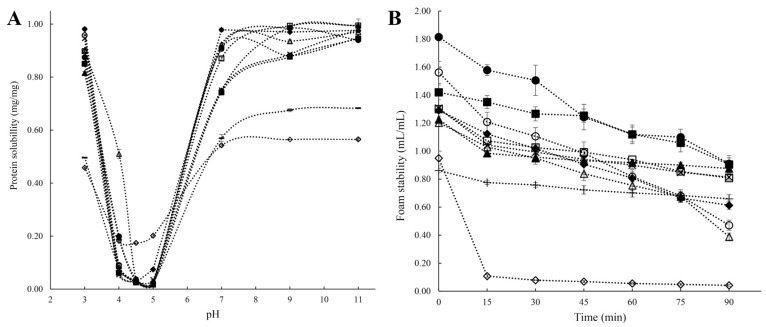
(**A**): Solubility of protein isolates under different pH; (**B**): Foam property (i.e., foam volume vs. time) of protein isolates. ■: White lentil protein isolate, □: Yellow lentil protein isolate, ●: Green mung bean protein isolate, ○: Yellow mung bean protein isolate, ◆: Soybean protein isolate, ◇: Commercial soybean protein isolate, ▲: Pigeon pea protein isolate, △: Cowpea protein isolate, ◊: Yellow pea protein isolate, -: Chickpea protein isolate [27]. Reproduced with permission from the copyright owner, published by Elsevier, 2021.

**Figure 4 foods-13-00006-f004:**
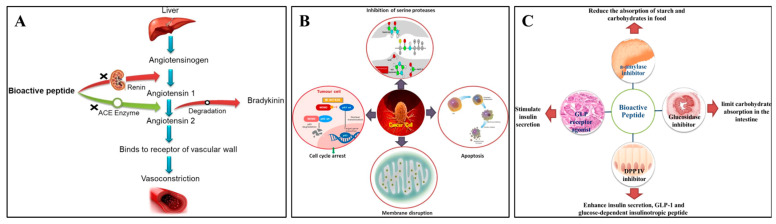
Antihypertensive mechanism (**A**), anticancer mechanism (**B**), and antidiabetic mechanism (**C**) of pulse bioactive peptides [78]. Reproduced with permission from the copyright owner, published by KLUWER ACADEMIC PUBLISHERS (DORDRECHT), 2021.

**Figure 5 foods-13-00006-f005:**
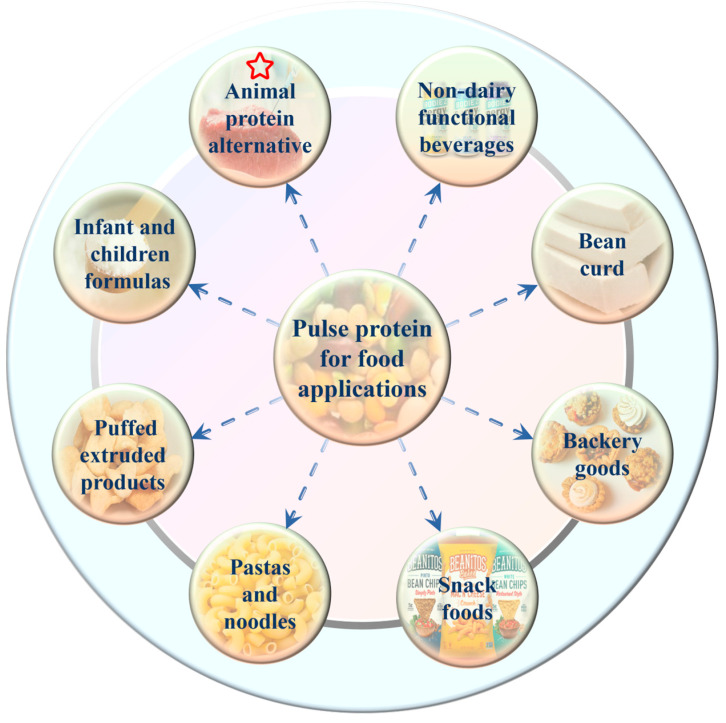
Main applications of pulse protein isolates in food industry [38].

**Figure 6 foods-13-00006-f006:**
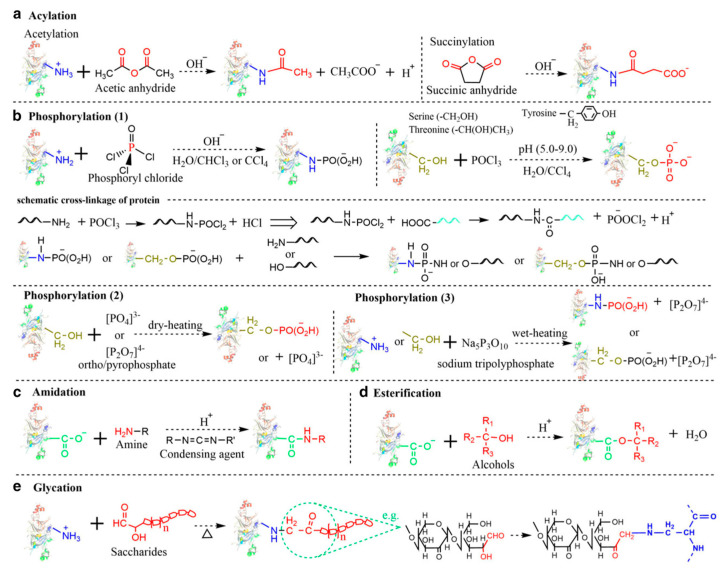
Simplified reaction process of various chemical modifications. (**a**) Acylation; (**b**) Phosphorylation; (**c**) Amidation; (**d**) Esterification; (**e**) Glycation (Maillard reaction) [9]. Reproduced with permission from the copyright owner, published by INSTITUTE OF FOOD TECHNOLOGISTS, 2021.

**Figure 7 foods-13-00006-f007:**
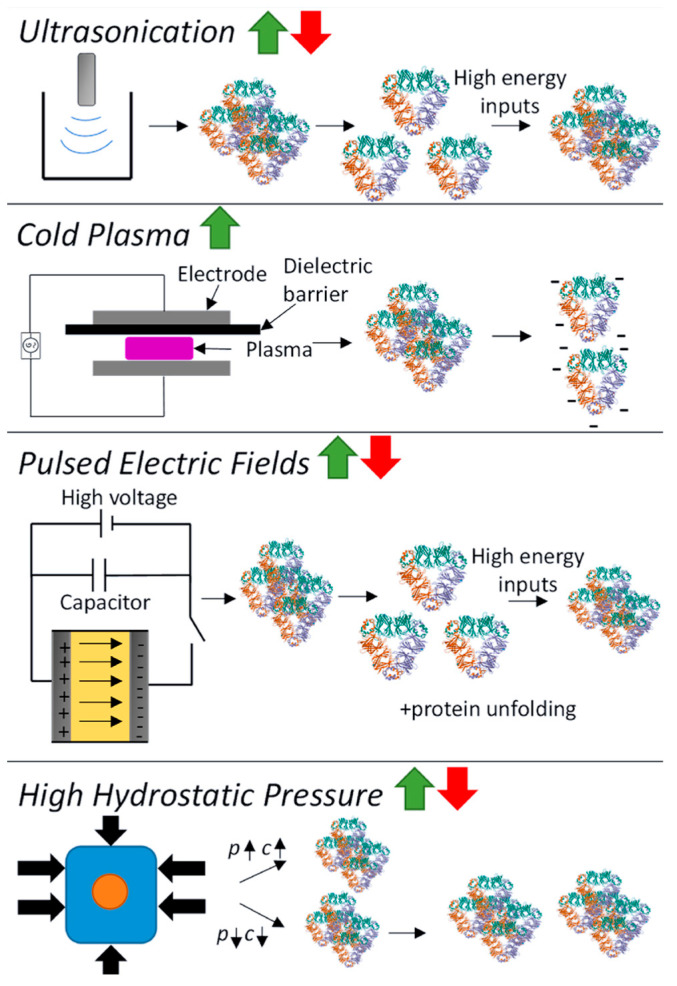
Schematic of the effect of physical modification (ultrasonication, cold plasma, pulsed electric fields, and high hydrostatic pressure) on pulse protein conformation. p = pressure, c = concentration [16]. Reproduced with permission from the copyright owner, published by Elsevier, 2023.

**Figure 8 foods-13-00006-f008:**
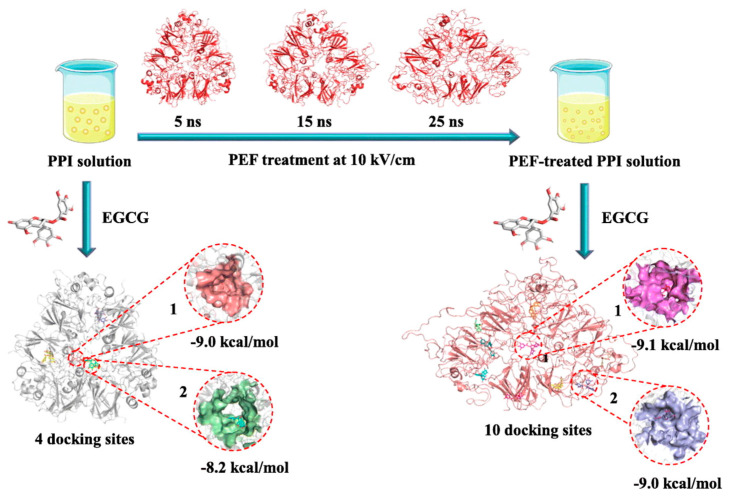
The schematic of pulsed electric field modification applied to enhance EGCG-binding capacity of pea protein isolate [111]. Reproduced with permission from the copyright owner, published by Elsevier, 2023.

**Table 3 foods-13-00006-t003:** Amino acid composition of pulse proteins (g/100 g dry matter) [9,11,25,27,37,38,39,40].

Amino Acid	Pea	Chickpea	Lentil	Mung Bean	Lupin	Cowpea	Faba Bean	Pigeon Pea
Essential AA								
Isoleucine (Ile, I)	0.4–4.9	0.4–4.1	0.5–5.0	1.0–4.7	1.2–3.2	4.3–4.4	1.1–4.3	4.8
Leucine (Leu, L)	1.3–8.4	0.5–7.0	0.8–7.9	1.8–8.4	2.0–7.4	7.1–7.5	2.0–8.2	7.5
Lysine (Lys, K)	1.4–7.7	0.9–7.7	0.5–7.2	1.7–4.2	1.2–7.6	3.9–6.6	1.9	4.4
Methionine (Met, M)	0.2–3.3	0.1–1.9	0.1–2.9	0.3–1.9	0.2–0.3	1.2–1.3	0.2–0.8	1.2
Phenylalanine (Phe, F)	0.2–8.1	0.4–5.9	0.6–7.8	1.1–5.7	1.0–3.3	4.0–5.6	1.2	3.9
Threonine (Thr, T)	0.9–4.5	0.1–3.6	0.6–3.8	0.8–3.2	1.0–4.3	2.5–3.7	1.0–13.0	2.8
Tryptophan (Trp, W)	0.2–1.0	0.2–1.1	0.7–0.8	0.3–1.0	0.2–0.3	0.3–1.1	0.2–1.1	NR
Valine (Val, V)	0.4–5.2	0.4–3.8	0.7–5.3	1.2–5.2	1.1–3.5	4.6–4.9	1.2	4.7
Arginine (Arg, R)	1.2–8.7	0.5–10.3	0.9–7.8	1.7–6.3	2.8–10.9	7.3	2.6–10.3	NR
Histidine (His, H)	0.5–2.8	0.2–3.4	0.4–3.4	0.7–3.6	0.7–3.1	2.8–3.5	0.9–2.7	4.0
Non-essential AA								
Alanine (Ala, A)	0.8–4.8	0.3–4.8	2.0–4.2	3.5–4.4	0.9–2.8	3.7–4.3	1.2–4.2	4.5
Aspartic acid (Asp, D)	2.1–11.9	0.6–11.4	1.1–11.3	8.4–13.5	2.8–8.4	7.8–11.9	3.1	8.2
Cystine (Cys, C)	0.4–1.6	1.3–2.3	0.0–1.0	0.8–1.8	0.3–0.6	1.0–1.8	0.4–1.9	2.2
Glutamic acid (Glu, E)	2.9–18.5	1.7–17.3	2.4–15.1	6.1–21.7	6.2–26.1	6.0–18.5	4.6–13.0	6.2
Glycine (Gly, G)	0.8–4.8	0.3–4.1	1.0–4.8	4.1–4.26	1.0–3.7	4.1–4.2	1.2–4.2	4.6
Proline (Pro, P)	0.8–4.6	0.2–4.6	0.9–3.8	2.8–4.2	1.1–4.3	2.8–3.6	1.2–3.9	3.0
Serine (Ser, S)	0.8–5.7	0.1–4.9	1.1–4.9	2.5–5.0	1.3–6.0	2.6–5.6	1.3	2.7
Tyrosine (Tyr, Y)	0.6–3.8	0.2–3.7	0.5–3.2	3.3–3.4	1.0–4.3	3.2–5.0	0.9	3.2

**Table 4 foods-13-00006-t004:** Osborne protein composition of pulse proteins (g/100 g dry matter).

	Albumin	Globulin	Glutelins	Prolamins
Pea [43,45]	18–25	55–65	3–4	4–5
Chickpea [44]	8–12	53–60	19–25	3–7
Lentil [39,46]	16–17	51–70	11	3–4
Mung bean [23,39]	16.3	62	13.3	0.9
Faba bean [47]	18.4–21.9	61.6–68	10.2–12.2	3.4–4.3
Cowpea [25]	4–12	58–80	10–15	1–3
Lupin [25]	9–22	44–60	6–23 a	

a = Sum of glutelin and prolamin.

## Data Availability

The data used to support the findings of this study can be made available by the corresponding author upon request.

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
