# Peer review of "Pulse Protein Isolates as Competitive Food Ingredients: Origin, Composition, Functionalities, and the State-of-the-Art Manufacturing"

_foods, 2023, doi:10.3390/foods13010006_

Round 1

Reviewer 1 Report

Comments and Suggestions for Authors

There many review articles concerning this topic with the same details, so what is your idea about the innovation and novelty of your repeated subject.

Pulse proteins: secondary structure, functionality and applications - PMC (nih.gov)

A comprehensive review on pulse protein fractionation and extraction: processes, functionality, and food applications: Critical Reviews in Food Science and Nutrition: Vol 0, No 0 (tandfonline.com)

Author Response

Reviewer #1

-There many review articles concerning this topic with the same details, so what is your idea about the innovation and novelty of your repeated subject.

-Pulse proteins: secondary structure, functionality, and applications - PMC (nih.gov)

-A comprehensive review on pulse protein fractionation and extraction: processes, functionality, and food applications: Critical Reviews in Food Science and Nutrition: Vol 0, No 0 (tandfonline.com)

Response:

Thank you for your valuable feedback and for taking the time to review our manuscript. We acknowledge the abundance of review articles on the topic and have carefully considered your concerns regarding the potential overlap with existing review articles, particularly the papers you provided.

In the first review paper, attention is directed towards elucidating the structural intricacies, functional attributes, and fundamental applications of pulse proteins up to the year 2019. And the second paper delves into contemporary methodologies pertaining to the isolation and extraction of pulse proteins. These two papers have been cited in our review. While previous reviews have covered certain aspects of pulse proteins and the core topic may be familiar, our manuscript uniquely emphasizes the advanced modification strategies of pulse proteins, particularly in the view of non-covalent complexation and physical modifications. And an up-to-date and comprehensive analysis was provided in this review, particularly in the context of emerging applications such as animal protein analogs. These unique perspectives are not extensively covered in the previously mentioned articles. What’s more, our manuscript synthesizes the latest research findings about pulse proteins, especially those published in 2022 and 2023, which may not have been covered in the earlier reviews. This ensures that our readers gain insights into the most recent advancements in the field of pulse protein.

Furthermore, I and other authors of this article have been actively involved in recent research on pulse proteins, such as the following articles: Teng, Y.X.; Zhang, T.; Dai, H.M.; Wang, Y.B.; Xu, J.T.; Zeng, X.A.; Li, B.; Zhu, X.W. Inducing the structural interplay of binary pulse protein complex to stimulate the solubilization of chickpea (Cicer arietinum L.) protein isolate. Food Chem. 2023, 407, 135136; Chen, Z.L.; Li, Y.; Wang, J.H.; Wang, R.; Teng, Y.X.; Lin, J.W.; Zeng, X.A.; Woo, M.W.; Wang, L.; Han, Z. Pulsed electric field improves the EGCG binding ability of pea protein isolate unraveled by multi-spectroscopy and computer simulation. Int. J. Biol. Macromol. 2023, 244, 125082; Rivera, J.; Siliveru, K.; Li, Y. A comprehensive review on pulse protein fractionation and extraction: processes, functionality, and food applications. Crit. Rev. Food Sci. Nutr. 2022, 1-23; Tang, X.; Shen, Y.T.; Zhang, Y.Q.; Schilling, M.W.; Li, Y.H. Parallel comparison of functional and physicochemical properties of common pulse proteins. LWT- Food Sci. Technol. 2021, 146, 111594; Shen, Y.; Hong, S.; Li, Y. Pea protein composition, functionality, modification, and food applications: A review. Adv. Food Nutr. Res. 2022, 101, 71-127.

Among them, the corresponding author of the second article you provided (A comprehensive review on pulse protein fractionation and extraction: processes, functionality, and food applications), Dr. Li, is also one of the authors of this review. Our most recent contributions have been consolidated and highlighted in this manuscript. To enhance the clarity of our contribution, the manuscript has been revised as below, explicitly emphasizing the novel aspects of our work.

Revisions made (Page 15-16, line 552 to 556): Also, Teng et al. designed a binary whole pulse protein complex when co-assembling pea protein with chickpea protein [15]. The novel binary protein presented superior solubility (50% higher than chickpea protein alone) due to the interplay between un-folded chickpea protein and pea protein during pH shifting, which enabled their resistance to acid-induced structural over-folding.

Revisions made (Page 17, line 600 to 604): As shown in Fig. 8, PEF treatment (10 kV/cm) enhanced the binding affinity of pea protein isolates with EGCG, increasing the binding constant by 2.35 times and binding sites from 4 to 10 [112]. The number of amino acid residues involved in hydrophobic interactions in PEF-treated pea protein increased from 5 to 13.

Revisions made (Page 18, line 645 to 648): Wang et al. prepared a chickpea protein hydrolysate by proteolytic hydrolysis and found that this hydrolysate had excellent cryoprotective effects on frozen surimi [121]. The chickpea protein hydrolysate alone (4%, w/w) exhibited comparable cryoprotective performance to that of the commercial formulation (4% sucrose and 4% sorbitol).

Revisions made (Page 19, line 672 to 696): Although many approaches have been developed and attempted to modify pulse protein isolates to improve their functionality in food products, a deep understanding of the composition and structure of pulse proteins is the key to further maximizing their utilization, as well as finding more approachable, scalable, and economic methods to increase their utilization. Till now the interfacial behaviors, gelation properties and hydration effects of most pulse proteins, including pea, beans, chickpea, and lentil, have been widely reported. Meanwhile, the obtained protein powders, hydrogels, or dispersions are playing an increasingly vital role in food formulations. It is also noteworthy that though the clean-label nature and health benefits of pulse proteins are obvious advantages over many other plant/animal resources, the functionalities of pulse proteins are still less competitive. For example, the poor solubility of chickpea protein hinders its use as an aqueous ingredient; the interfacial stability of pea protein is still worth further improvement compared with that of soybean. Thus, to realize more practical uses of pulse proteins in foods, modification is necessary. And the diversified origins and protein structures of pulse proteins also lead to the distinguished physicochemical properties among different pulse protein isolates. Developing effective and efficient screening mechanisms and workflows is essential considering the diversity of pulse protein products. To address these challenges, research in matching a specific protein candidate for certain food application can be critical in filling the gap, for ex-ample, i.e., chickpea for cryoprotectants, lenticel for binders, pea for interfacial stabilizers, etc., respectively. Moreover, novel, and green modification strategies for pulse proteins are still highly desirable, in which non-covalent multicomponent-based complexation, physical field-assisted modification and enzymatic modifications hold promise. These three aspects of modification strategies, along with suitable applications, of pulse proteins merit further investigation and are worthy of a more in-depth review discussion.

Reviewer 2 Report

Comments and Suggestions for Authors

Pulse protein isolates as competitive food ingredients: origin, composition, functionalities, and the state-of-the-art manufac-turing

It is a good investigation, however minor corrections must be made.

MInor concems:

1.    You can consult these articles to have more information to support your article.

Shrestha, S., Van t Hag, L., Haritos, V., Dhital, S. (2023). Comparative study on molecular and higher-order structures of legume seed protein isolates: Lentil, mungbean and yellow pea, Food Chemistry, 411: 1-12.

Jiao Ge, J., Sun, Cui.Xia., Fang, Y. (2022). Introducción del aislado de proteína de frijol panda (Vigna umbellata (Thunb.) Ohwi et Ohashi) como fuente alternativa de proteína de leguminosas: características fisicoquímicas, funcionales y nutricionales. Food Chemistry.

2.    conduct a broader review of the topic.

3.    Realizar una discusión más amplia sobre las conclusiones.

      4, Search for more recent research.

Comments on the Quality of English Language

Minor editing of English language required

Author Response

Reviewer #2

- It is a good investigation; however minor corrections must be made.

  1. You can consult these articles to have more information to support your article.

-Shrestha, S., Van t Hag, L., Haritos, V., Dhital, S. (2023). Comparative study on molecular and higher-order structures of legume seed protein isolates: Lentil, mungbean and yellow pea, Food Chemistry, 411: 1-12.

-Jiao Ge, J., Sun, Cui.Xia., Fang, Y. (2022). Introducción del aislado de proteína de frijol panda (Vigna umbellata (Thunb.) Ohwi et Ohashi) como fuente alternativa de proteína de leguminosas: características fisicoquímicas, funcionales y nutricionales. Food Chemistry.

Response: Thank you for your positive feedback. We appreciate your suggestions for additional references to strengthen the support for our article. After a thorough review of the recommended articles, the relevant information has been incorporated from them to enhance the comprehensiveness of our manuscript.

Revisions made (Page 5, line 169 to 171): Ge et al. found that panda bean (Vigna umbellate (Thunb.) Ohwi et Ohashi) protein isolate presents excellent amino-acid composition and protein efficiency ratio [40].

Revisions made (Page 7, line 228 to 232): Shrestha et al. identified the protein fraction composition of lentils and yellow pea based on molecular weight estimates from sodium dodecyl sulfate polyacrylamide gel electrophoresis analysis [54]. They found that soluble proteins from them were identi-fied as legume-like and vicilin-like, whereas vicilin-like proteins predominated in mung bean.

Revisions made (Page 11, line 370 to 372): Ge et al. reported that panda bean protein isolates presented superior emulsifying and foaming abilities, compared to soy protein isolates and pea protein isolates [40].

  1. Conduct a broader review of the topic.

Response: Thank you for your valuable suggestion. We acknowledge the importance of providing a comprehensive paper review. In response to your suggestion, more extensive reviews of the existing research have been conducted to ensure a thorough understanding of the topic. Additional relevant studies have been included, and the scope has been expanded to provide a more comprehensive overview.

Revisions made (Page 15-16, line 552 to 556): Also, Teng et al. designed a binary whole pulse protein complex when co-assembling pea protein with chickpea protein [15]. The novel binary protein presented superior solubility (50% higher than chickpea protein alone) due to the interplay between un-folded chickpea protein and pea protein during pH shifting, which enabled their resistance to acid-induced structural over-folding.

Revisions made (Page 17, line 600 to 604): As shown in Fig. 8, PEF treatment (10 kV/cm) enhanced the binding affinity of pea protein isolates with EGCG, increasing the binding constant by 2.35 times and binding sites from 4 to 10 [112]. The number of amino acid residues involved in hydrophobic interactions in PEF-treated pea protein increased from 5 to 13.

Revisions made (Page 18, line 645 to 648): Wang et al. prepared a chickpea protein hydrolysate by proteolytic hydrolysis and found that this hydrolysate had excellent cryoprotective effects on frozen surimi [121]. The chickpea protein hydrolysate alone (4%, w/w) exhibited comparable cryoprotective performance to that of the commercial formulation (4% sucrose and 4% sorbitol).

  1. Realizar una discusión más amplia sobre las conclusiones.

Response: Thank you for your constructive comments. The conclusion section has been revised to be broader and more enriching. Additional details on potentially efficient modification strategies and advanced applications of pulse proteins have been included, along with possible directions for future research.

Revisions made (Page 19, line 672 to 696): Although many approaches have been developed and attempted to modify pulse protein isolates to improve their functionality in food products, a deep understanding of the composition and structure of pulse proteins is the key to further maximizing their utilization, as well as finding more approachable, scalable, and economic methods to increase their utilization. Till now the interfacial behaviors, gelation properties and hydration effects of most pulse proteins, including pea, beans, chickpea, and lentil, have been widely reported. Meanwhile, the obtained protein powders, hydrogels, or dispersions are playing an increasingly vital role in food formulations. It is also noteworthy that though the clean-label nature and health benefits of pulse proteins are obvious advantages over many other plant/animal resources, the functionalities of pulse proteins are still less competitive. For example, the poor solubility of chickpea protein hinders its use as an aqueous ingredient; the interfacial stability of pea protein is still worth further improvement compared with that of soybean. Thus, to realize more practical uses of pulse proteins in foods, modification is necessary. And the diversified origins and protein structures of pulse proteins also lead to the distinguished physicochemical properties among different pulse protein isolates. Developing effective and efficient screening mechanisms and workflows is essential considering the diversity of pulse protein products. To address these challenges, research in matching a specific protein candidate for certain food application can be critical in filling the gap, for ex-ample, i.e., chickpea for cryoprotectants, lenticel for binders, pea for interfacial stabilizers, etc., respectively. Moreover, novel, and green modification strategies for pulse proteins are still highly desirable, in which non-covalent multicomponent-based complexation, physical field-assisted modification and enzymatic modifications hold promise. These three aspects of modification strategies, along with suitable applications, of pulse proteins merit further investigation and are worthy of a more in-depth review discussion.

  1. Search for more recent research.

Response: Thank you for your valuable suggestion. Our manuscript has further summarized numerous articles published on pulse proteins in recent years.

Revisions made (Page 15-16, line 552 to 556): Also, Teng et al. designed a binary whole pulse protein complex when co-assembling pea protein with chickpea protein [15]. The novel binary protein presented superior solubility (50% higher than chickpea protein alone) due to the interplay between un-folded chickpea protein and pea protein during pH shifting, which enabled their resistance to acid-induced structural over-folding.

Revisions made (Page 17, line 600 to 604): As shown in Fig. 8, PEF treatment (10 kV/cm) enhanced the binding affinity of pea protein isolates with EGCG, increasing the binding constant by 2.35 times and binding sites from 4 to 10 [112]. The number of amino acid residues involved in hydrophobic interactions in PEF-treated pea protein increased from 5 to 13.

Revisions made (Page 18, line 645 to 648): Wang et al. prepared a chickpea protein hydrolysate by proteolytic hydrolysis and found that this hydrolysate had excellent cryoprotective effects on frozen surimi [121]. The chickpea protein hydrolysate alone (4%, w/w) exhibited comparable cryoprotective performance to that of the commercial formulation (4% sucrose and 4% sorbitol).

Reviewer 3 Report

Comments and Suggestions for Authors

The content of the manuscript is interesting, but it lacks the key elements of a scientific manuscript:

The manuscript is not written according to the IMRAD scheme.

The methods chapter on a comprehensive review, i.e. a systematic, scientifically designed review of a defined literature base that uses the rigour of original research to limit bias in the results, is missing. It should include an explanation of the inclusion and exclusion criteria for the studies included in the review; the literature search should be defined and applied before the literature search begins.

A systematic presentation of the results is also missing.

There is a lack of detailed information on the limitations identified in the review.

Author Response

Reviewer #3

- The content of the manuscript is interesting, but it lacks the key elements of a scientific manuscript.

  1. The manuscript is not written according to the IMRAD scheme.

Response:

Thank you for your thoughtful review. We acknowledge the oversight in not strictly adhering to the IMRAD (Introduction, Methods, Results, and Discussion) scheme. For writing SCI papers, the IMRAD writing structure is very scientific and well suited for research papers.

To enhance the framework of our paper, we examined several review articles recently published in the Foods, such as “Yao, J.; Chen, W.; Fan, K. Novel Efficient Physical Technologies for Enhancing Freeze Drying of Fruits and Vegetables: A Review. Foods 2023, 12, 4321. https://doi.org/10.3390/foods12234321”, “Ali, S.; Freire, L.G.D.; Rezende, V.T.; Noman, M.; Ullah, S.; Abdullah; Badshah, G.; Afridi, M.S.; Tonin, F.G.; de Oliveira, C.A.F. Occurrence of Mycotoxins in Foods: Unraveling the Knowledge Gaps on Their Persistence in Food Production Systems. Foods 2023, 12, 4314. https://doi.org/10.3390/foods12234314”. In fact, the structure of these papers did not strictly adhere to the IMRAD scheme. Furthermore, we meticulously reviewed the author guidelines of the Foods regarding the requirements for review articles, which are outlined as follows: “Review manuscripts should comprise the front matter, literature review sections and the back matter. The template file can also be used to prepare the front and back matter of your review manuscript. It is not necessary to follow the remaining structure.”

Therefore, considering both the original logical structure of our manuscript and the diverse structural preferences outlined by the Foods for review articles, it has been decided to retain the original framework of this review paper. In order to enhance the scientific rigor and readability of the article, the manuscript has been revised as below, further emphasizing the theme of modified strategies for pulse proteins and their cutting-edge applications in the food industry.

Revisions made (Page 15-16, line 552 to 556): Also, Teng et al. designed a binary whole pulse protein complex when co-assembling pea protein with chickpea protein [15]. The novel binary protein presented superior solubility (50% higher than chickpea protein alone) due to the interplay between un-folded chickpea protein and pea protein during pH shifting, which enabled their resistance to acid-induced structural over-folding.

Revisions made (Page 17, line 600 to 604): As shown in Fig. 8, PEF treatment (10 kV/cm) enhanced the binding affinity of pea protein isolates with EGCG, increasing the binding constant by 2.35 times and binding sites from 4 to 10 [112]. The number of amino acid residues involved in hydrophobic interactions in PEF-treated pea protein increased from 5 to 13.

Revisions made (Page 18, line 645 to 648): Wang et al. prepared a chickpea protein hydrolysate by proteolytic hydrolysis and found that this hydrolysate had excellent cryoprotective effects on frozen surimi [121]. The chickpea protein hydrolysate alone (4%, w/w) exhibited comparable cryoprotective performance to that of the commercial formulation (4% sucrose and 4% sorbitol).

Revisions made (Page 19, line 672 to 696): Although many approaches have been developed and attempted to modify pulse protein isolates to improve their functionality in food products, a deep understanding of the composition and structure of pulse proteins is the key to further maximizing their utilization, as well as finding more approachable, scalable, and economic methods to increase their utilization. Till now the interfacial behaviors, gelation properties and hydration effects of most pulse proteins, including pea, beans, chickpea, and lentil, have been widely reported. Meanwhile, the obtained protein powders, hydrogels, or dispersions are playing an increasingly vital role in food formulations. It is also noteworthy that though the clean-label nature and health benefits of pulse proteins are obvious advantages over many other plant/animal resources, the functionalities of pulse proteins are still less competitive. For example, the poor solubility of chickpea protein hinders its use as an aqueous ingredient; the interfacial stability of pea protein is still worth further improvement compared with that of soybean. Thus, to realize more practical uses of pulse proteins in foods, modification is necessary. And the diversified origins and protein structures of pulse proteins also lead to the distinguished physicochemical properties among different pulse protein isolates. Developing effective and efficient screening mechanisms and workflows is essential considering the diversity of pulse protein products. To address these challenges, research in matching a specific protein candidate for certain food application can be critical in filling the gap, for ex-ample, i.e., chickpea for cryoprotectants, lenticel for binders, pea for interfacial stabilizers, etc., respectively. Moreover, novel, and green modification strategies for pulse proteins are still highly desirable, in which non-covalent multicomponent-based complexation, physical field-assisted modification and enzymatic modifications hold promise. These three aspects of modification strategies, along with suitable applications, of pulse proteins merit further investigation and are worthy of a more in-depth review discussion.

  1. The methods chapter on a comprehensive review, i.e., a systematic, scientifically designed review of a defined literature base that uses the rigour of original research to limit bias in the results, is missing. It should include an explanation of the inclusion and exclusion criteria for the studies included in the review; the literature search should be defined and applied before the literature search begins. A systematic presentation of the results is also missing.

Response: Thank for your feedback. Your point regarding the lack of a systematic presentation of Methods section and Results section is noted. However, as mentioned in last response, after considering various factors comprehensively, this manuscript has not been structured according to the IMRAD (Introduction, Methods, Results, and Discussion) format.

  1. There is a lack of detailed information on the limitations identified in the review.

Response: Thank you for your valuable suggestion. The lack of detailed information on limitations identified in the review has been addressed by explicitly outlining the limitations of our review in the Conclusion section. We hope that these proposed revisions will address the deficiencies you highlighted and elevate the manuscript to meet the rigorous standards expected for Foods publications.

Revisions made (Page 19, line 672 to 696): Although many approaches have been developed and attempted to modify pulse protein isolates to improve their functionality in food products, a deep understanding of the composition and structure of pulse proteins is the key to further maximizing their utilization, as well as finding more approachable, scalable, and economic methods to increase their utilization. Till now the interfacial behaviors, gelation properties and hydration effects of most pulse proteins, including pea, beans, chickpea, and lentil, have been widely reported. Meanwhile, the obtained protein powders, hydrogels, or dispersions are playing an increasingly vital role in food formulations. It is also noteworthy that though the clean-label nature and health benefits of pulse proteins are obvious advantages over many other plant/animal resources, the functionalities of pulse proteins are still less competitive. For example, the poor solubility of chickpea protein hinders its use as an aqueous ingredient; the interfacial stability of pea protein is still worth further improvement compared with that of soybean. Thus, to realize more practical uses of pulse proteins in foods, modification is necessary. And the diversified origins and protein structures of pulse proteins also lead to the distinguished physicochemical properties among different pulse protein isolates. Developing effective and efficient screening mechanisms and workflows is essential considering the diversity of pulse protein products. To address these challenges, research in matching a specific protein candidate for certain food application can be critical in filling the gap, for ex-ample, i.e., chickpea for cryoprotectants, lenticel for binders, pea for interfacial stabilizers, etc., respectively. Moreover, novel, and green modification strategies for pulse proteins are still highly desirable, in which non-covalent multicomponent-based complexation, physical field-assisted modification and enzymatic modifications hold promise. These three aspects of modification strategies, along with suitable applications, of pulse proteins merit further investigation and are worthy of a more in-depth review discussion.

Round 2

Reviewer 1 Report

Comments and Suggestions for Authors

I would like to appreciate the author strategy to defend his topic as well as his response to my claim. The author clarification may highlight the small difference between his study and other previous familiar ones.

Author Response

Thank you for your valuable feedback and effort on our manuscript. Your positive acknowledgment of our revision to defend the topic and respond to your claim is encouraging. Once again, thank you for your recognition and thoughtful engagement with our work.

Reviewer 3 Report

Comments and Suggestions for Authors

Thank you for the answer, but unfortunately it is not satisfactory. If the article is also a review, it must include a described methodology or at least a description of how you arrived at the results, and systematic reviews should follow the PRISMA guidelines.

Author Response

Response: Thank you for your continued engagement with our manuscript and for providing additional guidance on the methodology section. We appreciate your thorough evaluation and understand the importance of a well-described methodology, particularly in the context of a review article. Consequently, we conducted a comprehensive review of the PRISMA guidelines and thoroughly examined various recent review articles [1–4] published in 'Foods' journal, with particular attention to their methodological sections. Subsequently, corresponding modifications have been made to our manuscript, incorporating detailed additions to the method section, mainly focusing on data sources, search strategy, and software used.

Revisions made (Page 3, line 98 to 128):

  1. Materials and Methods

2.1. Data Sources

A thorough search was conducted to gather studies investigating strategies aimed at enhancing the functional properties of pulse proteins and broadening their applications in the food industry. The systematic review focused on published articles in the English language, excluding reviews, spanning the years 2010 to 2023, i.e., with a specific emphasis on the last three years. The databases utilized for collecting relevant articles were Web of Science (https://webofscience.clarivate.cn/wos/alldb/basic-search, accessed on July 15, 2023) and Elsevier (https://www.sciencedirect.com/, accessed on July 15, 2023). Additionally, partial worldwide data on pulse proteins and protein crystal structures were obtained from Our World in Data (https://ourworldindata.org/) and the PDB database (http://www.rcsb.org/pdb/), respectively.

2.2. Search Strategy

The search strategy employed the following keywords: (pulse crops OR pulse protein isolates OR pea protein OR chickpea protein OR cowpea protein OR lentils protein OR bean proteins OR faba bean protein OR mung bean protein) AND (solubility OR water holding capacity OR oil holding capacity OR emulsifying properties OR foaming properties OR gelation properties OR bioactive properties) AND (chemical modification OR complexation OR interaction OR physical modification OR biological modification) OR interaction OR [food application]. All the initial literature records were exported in full-record format. Following this, through a meticulous review of the complete texts, studies deemed irrelevant were systematically excluded. Relevance assessments were conducted by all authors, and consensus was reached through collaborative evaluation. Ultimately, a total of 473 articles that met the established criteria were retained for further in-depth analysis.

2.3. Software Used

Within the scope of this work, the following software applications were employed: Origin 2023 (OriginLab, Northampton, MA, USA), PowerPoint 365 (Microsoft, Bellvue, MA, USA), and Photoshop 2023 (Adobe Systems Incorporated, San Jose, CA, USA) for the generation of visual representations. Data processing was conducted using Microsoft Office 365 Excel (Microsoft, Bellvue, MA, USA).

Reference in response letter

  1. Akbar, J.; Gul, M.; Jahangir, M.; Adnan, M.; Saud, S.; Hassan, S.; Nawaz, T.; Fahad, S. Global Trends in Halal Food Standards: A Review. Foods 2023, 12, 4200.
  2. Bueno, C.; Thys, R.; Tischer, B. Potential Effects of the Different Matrices to Enhance the Polyphenolic Content and Antioxidant Activity in Gluten-Free Bread. Foods 2023, 12, 4415.
  3. Deligeorgakis, C.; Magro, C.; Skendi, A.; Gebrehiwot, H.H.; Valdramidis, V.; Papageorgiou, M. Fungal and Toxin Contaminants in Cereal Grains and Flours: Systematic Review and Meta-Analysis. Foods 2023, 12, 4328.
  4. Zhang, J.; Wei, Z.; Lu, T.; Qi, X.; Xie, L.; Vincenzetti, S.; Polidori, P.; Li, L.; Liu, G. The Research Field of Meat Preservation: A Scientometric and Visualization Analysis Based on the Web of Science. Foods 2023, 12, 4239.